# Auto-Connect: Connectivity-Preserving RigFormer with Direct Preference Optimization

**Jingfeng Guo**[1*], **Jian Liu**[2,7*], **Jinnan Chen**[3†], **Shiwei Mao**[4,7], **Changrong Hu**[5,7], **Puhua Jiang**[4,7]

**Junlin Yu**[6,7], **Jing Xu**[7], **Qi Liu**[1†], **Lixin Xu**[7], **Zhuo Chen**[7], **Chunchao Guo**[7]

[1]South China University of Technology

[2]Hong Kong University of Science and Technology [3]National University of Singapore

[4]Tsinghua Shenzhen International Graduate School [5]University of Science and Technology of China

[6]Beijing Normal University [7]Tencent Hunyuan

https://autoconnectrig.github.io/

## Abstract

We introduce Auto-Connect, a novel approach for automatic rigging that explicitly preserves skeletal connectivity through a connectivity-preserving tokenization scheme. Unlike previous methods that predict bone positions represented as two joints or first predict points before determining connectivity, our method employs special tokens to define endpoints for each joint's children and for each hierarchical layer, effectively automating connectivity relationships. This approach significantly enhances topological accuracy by integrating connectivity information directly into the prediction framework. To further guarantee high-quality topology, we implement a topology-aware reward function that quantifies topological correctness, which is then utilized in a post-training phase through reward-guided Direct Preference Optimization. Additionally, we incorporate implicit geodesic features for latent top-$k$ bone selection, which substantially improves skinning quality. By leveraging geodesic distance information within the model's latent space, our approach intelligently determines the most influential bones for each vertex, effectively mitigating common skinning artifacts. This combination of connectivity-preserving tokenization, reward-guided fine-tuning, and geodesic-aware bone selection enables our model to consistently generate more anatomically plausible skeletal structures with superior deformation properties.

## 1 Introduction

The creation of highly detailed 3D shapes [1, 2, 3, 4, 5, 6, 7, 8, 9, 10, 11] and digital avatars [12, 13, 14, 15, 16, 17, 18, 19, 20, 21, 22] has become increasingly accessible through advancements in generative modeling technologies. Despite these impressive capabilities in static content generation, these models remain fundamentally limited for dynamic applications. Some early works address this limitation by predicting dynamics through per-vertex deformation or physical simulation [23, 24, 25, 26, 27, 28, 29, 30, 31]. In contrast, rigging offers a parametric representation that establishes a skeleton for a 3D model and defines surface deformation in response to skeletal movement which is compatible to graphics pipeline. Despite its critical importance in animation pipelines, auto-rigging

---

[*]Equal contributions. Work primarily done during an internship at Tencent Hunyuan

[†]Corresponding author.

39th Conference on Neural Information Processing Systems (NeurIPS 2025).

remains a challenging problem due to the complexity of accurately modeling skeletal structures, ensuring proper joint and bone connectivity, and preventing artifacts during animation.

Early approaches to auto-rigging relied on fixed topology templates [32, 25], which limited their applicability across diverse character morphologies. Later developments, such as RigNet [33], employed more flexible strategies including clustering for joint position acquisition and Minimum Spanning Tree (MST) algorithms for topology construction. Additionally, various optimization-based methods [34, 35, 36, 37] have been proposed to generate character-specific rigs, but often require additional optimization cost and suffer from generalization issues. Recent advancements in generative models and the expansion of 3D datasets have made auto-rigging more scalable.

Current learning-based approaches typically fall into two categories: methods [38, 39] that predict bone positions without explicitly considering the connectivity between joints, and methods [40] that first predict skeletal joint points before determining their connectivity. As a result, these approaches frequently produce suboptimal topology quality, primarily because their representations fail to effectively capture the inherent topological relationships within skeletal structures. This deficiency frequently leads to unrealistic deformations in animations.

To address these limitations, we introduce Auto-Connect, a novel approach that explicitly preserves skeletal connectivity. By incorporating a connectivity-preserving tokenization scheme, Auto-Connect ensures that joints and bones are connected in a way that maintains the inherent structure of the skeleton, overcoming the topological errors common in previous methods. Our method defines endpoints for each joint's children and for each hierarchical layer, effectively automating connectivity relationships within the prediction framework itself. This fundamental redesign significantly enhances topological accuracy by integrating connectivity information directly into the model's representation. Based on this representation, we design an autoregressive training framework for skeleton trees, RigFormer, which incorporates level position embedding and a set of data augmentation strategies.

Furthermore, purely next-token prediction with cross-entropy loss focuses solely on local conditional distribution modeling, failing to adequately capture joint distribution properties. To address this limitation, we introduce a joint distribution constraint during post-training through a DPO loss framework. Our approach primarily emphasizes topology improvement by incorporating carefully designed topology-aware reward functions. We evaluate rig quality based on joint position accuracy and topological quality. For position accuracy, we calculate the chamfer distance between predicted joint positions and ground truth, providing a robust measure of geometric fidelity. For topological quality, we employ two complementary metrics: Tree Edit Distance, which measures the cost of transforming the predicted skeleton topology into the ground truth through a series of edit operations, and Hierarchical Jaccard Similarity, which quantifies the overlap between hierarchical structures while considering parent-child relationships. This reward function is then utilized in a post-training phase through our reward-guided Direct Preference Optimization (DPO), guiding the model toward generating both geometrically accurate and topologically sound skeletal structures. Finally, we incorporate implicit geodesic features for latent top-$k$ bone selection, which substantially improves skinning quality by leveraging spatial relationships within the character's geometry. This approach implicitly determines the most influential bones for each vertex, effectively mitigating common skinning artifacts, particularly stretching phenomena that occur during extreme deformations.

Through extensive experiments on public benchmarks, we demonstrate that Auto-Connect substantially outperforms previous state-of-the-art methods, achieving superior results in joint location accuracy, topological consistency, and skinning quality. We summarize our contributions as follows:

- We introduce Auto-Connect, a novel automatic rigging pipeline that explicitly preserves skeletal connectivity through a connectivity-preserving tokenization scheme coupled with an enhanced pre-training framework, RigFormer.

- We develop a topology-aware reward function tailored for skeleton tree structures, and build upon this, we present a rigging post-training phase through our reward-guided DPO to further improve topology quality. To the best of our knowledge, this is the first work to combine reinforcement learning with the rigging task.

- We present a plug-and-play geodesic-aware bone probability prediction module that incorporates implicit geodesic features to dynamically determine the top-$k$ bone for each vertex, effectively mitigating common skinning artifacts.

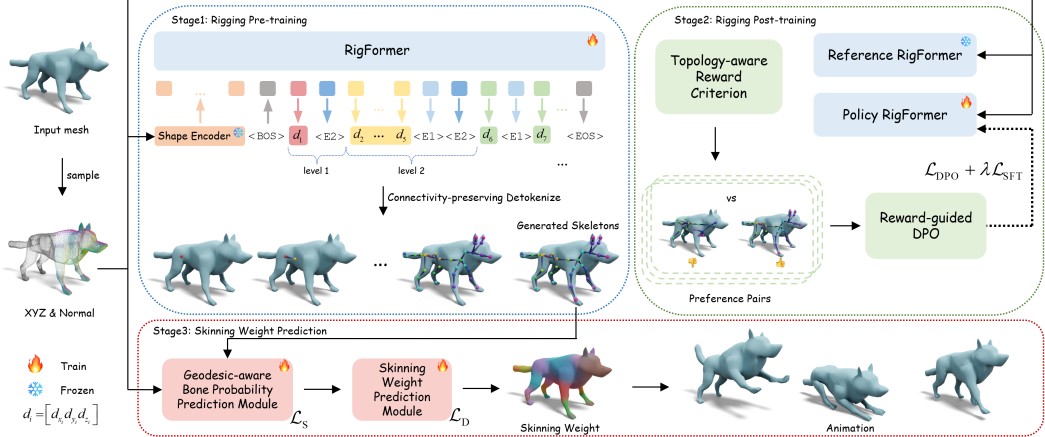

Figure 1: **Overview of the Auto-Connect.** The pipeline consists of three main stages. In the **Rigging Pre-training** stage, a point cloud sampled from the input 3D mesh is processed by the shape encoder to extract geometric features, which are subsequently fed into our autoregressive RigFormer to generate a token sequence. The generated sequence is then processed using our connectivity-preserving detokenization to construct the skeleton tree. In the **Rigging Post-training** stage, preference pairs are constructed using our topology-aware reward criterion, and RigFormer is fine-tuned with our reward-guided DPO for preference-driven optimization. Finally, in the **Skinning Weight Prediction** stage, the generated skeleton and mesh vertices serve as input. Our geodesic-aware bone probability prediction module is employed to implicitly determine the most influential bones to predict the skinning weights, enabling mesh animation.

## 2 Related Work

### 2.1 Automatic Rigging and Skinning

Automatic Rigging can be categorized into template-based [32, 41, 42, 43, 44] methods and template-free [33, 45, 40, 38, 39] paradigms. Template-based methods mainly focus on humanoid characters and are limited to fixed skeleton topologies, restricting their use for diverse character types. Template-free methods like RigNet [33] and AnimSkelVolNet [45] use regression and clustering for joint prediction, along with MST for connectivity. However, their hybrid architectures face optimization challenges due to non-differentiable clustering and MST operations. RigAnything [40] and MagicArticulate [38] use autoregressive rigging but neglect the skeleton's hierarchical structure and parent-child joint connections. This forces them to either rely on additional modules for joint connectivity or re-encode parent joints multiple times, which increases sequence length and computational cost. UniRig [39] attempts to address these shortcomings by extracting bone chains through Depth-First Search. However, the connections between bone chains rely on heuristic connection rules, which merge joints within a predefined distance threshold, making it non-end-to-end and prone to compounding errors. Additionally, it often predicts the termination token prematurely, leading to incomplete skeletal structures with missing bone chains. In contrast, our method overcomes these limitations by integrating topology information directly into the model's representation. Current skinning methods [33, 46, 47] statically select the $k$-nearest bones for each vertex and assume that only these bones influence the vertex, then use Graph Neural Networks to predict skinning weights. However, complex mesh-skeleton topologies often render distance calculations unreliable, leading to critical binding errors. To address this, we present a plug-and-play geodesic-aware module that dynamically identify the $k$ most probable influencing bones conditioned on geodesic feature cues.

### 2.2 Autoregressive Models for 3D Generation

Autoregressive transformers [48, 49] have radically transformed visual generation [50, 51, 52, 53, 54, 55] through their sophisticated sequential approach of synthesizing images using discrete tokens derived from image tokenizers. This paradigm has achieved remarkable success by decomposing the complex image generation task into a series of manageable token prediction steps, enabling more coherent and controllable outputs. Building upon this foundation, recent work [4, 56, 57, 58, 59, 60, 61] has introduced specialized mesh tokenizers that extend the autoregressive framework to 3D mesh

generation. These methods effectively discretize 3D geometry into sequential tokens that can be predicted in an autoregressive manner, similar to language modeling. Our method builds upon these advances, introducing a novel connectivity-preserving tokenizer that enables more accurate, diverse, and artist-intuitive skeleton generations.

## 2.3 RLHF with Direct Preference Optimization

With the rapid advancement of Large Language Models (LLMs) [62, 63] and Vision-Language Models (VLMs) [64, 65, 66], aligning policy models with human preferences has become increasingly critical. Post-training techniques such as Reinforcement Learning with Human Feedback (RLHF) and Direct Preference Optimization (DPO) aim to improve model performance by reflecting user intentions. Early RLHF methods [67] trained on manually labeled preference pairs and optimized policies using Proximal Policy Optimization (PPO) [68], but faced instability and high computational costs. DPO addresses this by removing the need for an explicit reward model and using an implicit reward function based on PPO optimality, optimizing policies via maximum likelihood estimation with the Bradley-Terry model [69]. Building upon this foundation, we design a topology-aware reward function for skeleton trees and propose a reward-guided DPO to encourage the generation of topologically accurate skeletal structures. To the best of our knowledge, this is the first successful implementation of DPO in rigging tasks.

## 3 Method

Our Auto-Connect comprises three core innovations, as illustrated in Figure 1. First, Section 3.1 introduces the rigging pre-training stage with a novel connectivity-preserving tokenization scheme and RigFormer training framework. Building upon this foundation, Section 3.2 presents the rigging post-training stage using our reward-guided DPO with the proposed topology-aware reward criteria. Finally, Section 3.3 details the proposed geodesic-aware bone probability prediction module that enables plug-and-play integration with existing skinning methods for precise deformation control.

## 3.1 Rigging Pre-training

Unlike template-based methods [41, 42, 43, 44] that rely on fixed topologies for generating specific skeleton categories, or prior template-free methods [33, 40, 38, 39] that overlook skeleton topology encoding, we propose a connectivity-preserving tokenizer that efficiently encodes both skeleton hierarchical structure and parent-child joint connections. This tokenizer enables the automation of joint connectivity in an autoregressive paradigm, allowing for the generation of diverse skeletons.

**Connectivity-preserving Tokenization.**
Given an input mesh and skeleton, we first normalize them into a unit cube space $[-0.5, 0.5]^3$ and apply n-bit quantization to the joint coordinates via

$$d_k = \lfloor (j_k + 0.5) \times 2^n \rfloor \, k \in \{x, y, z\} \quad (1)$$

where $j_k$ and $d_k$ denote the original and discretized coordinates, respectively. Next, we traverse the skeleton tree in a breadth-first (BFS) order, as shown in Figure 2. Specifically, based on the standard BFS traversal, we insert a special token <E1> after visiting all the child joints of a parent joint to indicate the endpoints for the

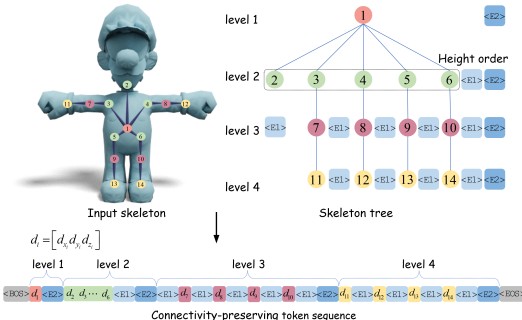

Figure 2: **Connectivity-preserving tokenization process.** The number indicates the joint indices.

current parent's children. Note that leaf joints not at the last level still need an <E1> to signify this. Similarly, after traversing all the joints in the current level, we insert another special token <E2> to mark the completion of that level. Additionally, we incorporate a height-aware spatial prior by sorting child joints under each parent joint according to their z-axis coordinates, which reduces the difficulty for the model in regressing joint positions. Finally, we obtain $3J + M + L$ tokens, where $J$ is the number of joints, $L$ is the number of levels in the skeleton tree, and $M$ is the total number of joints in the first $L - 1$ levels.

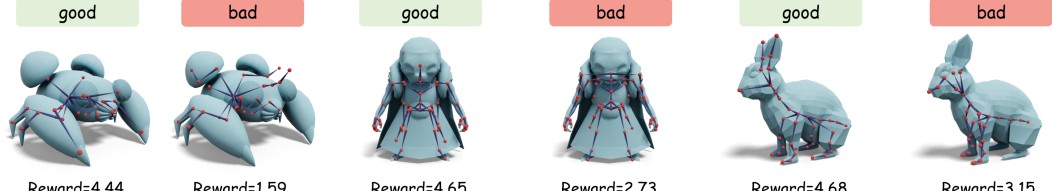

Figure 3: **Examples of the collected preference pairs.** Skeleton trees with higher reward exhibit superior topology and better align with human intuition, making them the preferred choice.

**Shape-conditioned generation.** we randomly sample $N = 20000$ surface points from the input mesh to construct a point cloud representation $\mathcal{P} \in \mathbb{R}^{N \times 3}$ with corresponding surface normals $\mathcal{N} \in \mathbb{R}^{N \times 3}$. These geometric primitives are encoded through a pre-trained Michelangelo [70] encoder $\mathcal{E}_g$, which captures both local geometric details and global shape semantics. The generates shape condition $\mathbf{c_g} = \mathcal{E}_g(\mathcal{P}, \mathcal{N})$, serving as the context for the autoregressive generation process.

**RigFormer.** We adopt a standard transformer with parameter $\theta$ to model the skeleton sequence and leverage cross-attention for shape conditioning. The training process is achieved using the next token prediction loss:

$$\mathcal{L}_{\text{stage1}} = -\prod_{i=1}^{T} p\left(\tau_i \,\middle|\, \tau_{1:i-1} \oplus \mathbf{e}_{\ell_{1:i-1}}, \mathbf{c}_g; \theta\right) \tag{2}$$

where $T$ denotes the total sequence length. To explicitly model skeletal hierarchy, we additionally inject level position embedding $\mathbf{e}_\ell$ into the transformer. Here, $p\left(\tau_i \,\middle|\, \tau_{1:i-1} \oplus \mathbf{e}_{\ell_{1:i-1}}\right)$ denotes the conditional probability of token $\tau_i$ given the preceding tokens and level embeddings in the sequence.

During inference, the generation process starts with only the shape tokens as input and progressively generates skeleton tokens. The resulting token sequence is then converted into the final skeleton using our connectivity-preserving detokenization. With the proposed special tokens <E1> and <E2>, the hierarchical structure and parent-child joint connections can be automatically determined.

## 3.2 Rigging Post-training

Next-token prediction method focuses only on local conditional distributions, neglecting the critical aspects of joint distribution modeling, especially for topology preserving. We implement a joint distribution constraint during the post-training phase, utilizing DPO loss to further improve the topology quality. Specifically, we introduce a topology-aware reward function to evaluate the quality of skeletons. Based on this criterion, we construct preference pairs and propose a reward-guided DPO to fine-tune our RigFormer in a preference-driven manner. Moreover, to prevent overfitting, we add the SFT auxiliary constraint loss during DPO training.

**Topology-aware Reward.** Given the predicted skeleton $\mathcal{S}_p$ and ground truth skeleton $\mathcal{S}_g$, we evaluate the quality of the predicted skeleton in terms of spatial accuracy and topological fidelity. Spatial accuracy is measured by the Chamfer Distance (CD) between the generated and ground truth skeletons, which is calculated as the sum of CD-J2J, CD-J2B, and CD-B2B. Each reward decays linearly from 1 to 0 points as the CD value increases from 0 to 10%, which can be formalized as:

$$R_{\text{CD}} = \sum_{i \in \{\text{J2J,J2B,B2B}\}} \max\left(1 - \frac{\text{CD}_i}{10\%}, 0\right) \in [0, 3] \tag{3}$$

For topological fidelity, we first use the Tree Edit Distance (TED) to measure the minimum edit operations required to transform the predicted skeleton into the ground truth skeleton. This value is normalized by the total number of joints in both trees to eliminate scale bias:

$$R_{\text{TED}} = 1 - \frac{\text{TED}\left(\mathcal{S}_p, \mathcal{S}_g\right)}{|\mathcal{J}_p| + |\mathcal{J}_g|}, R_{\text{TED}} \in [0, 1] \tag{4}$$

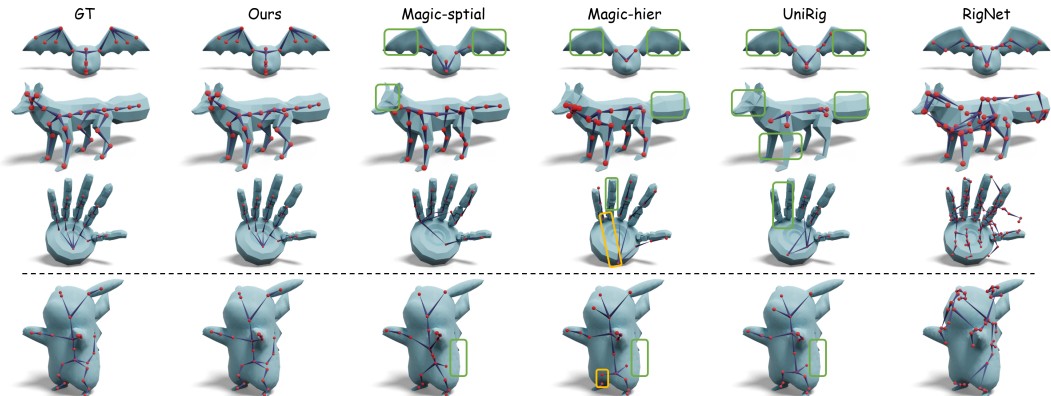

Figure 4: **Qualitative comparison of rigging result on Art-XL2.0 (top) and MR (bottom).** Our connectivity-preserving representation effectively captures intrinsic skeletal topology, and reward-guided fine-tuning enables the generated skeletons to better align with artistic aesthetics. Additional results are provided in the appendix B.1.

where $|\mathcal{J}_p|$ and $|\mathcal{J}_g|$ represent the number of joints. Then, we calculate the Hierarchical Jaccard Similarity (HJS), which evaluates the local topological accuracy of the shared joints $\mathcal{J}_{\text{common}}$:

$$R_{\text{HJS}} = \frac{1}{|\mathcal{J}_{\text{common}}|} \sum_{j \in \mathcal{J}_{\text{common}}} \frac{\left| \mathcal{C}_{p(j)} \cap \mathcal{C}_{g(j)} \right|}{\left| \mathcal{C}_{p(j)} \cup \mathcal{C}_{g(j)} \right|} \in [0, 1] \tag{5}$$

where $\mathcal{C}_{p(j)}$ and $\mathcal{C}_{g(j)}$ represent the children set of joint $j$ in the predicted and ground truth skeletons respectively. The composite reward aggregates these as a 5-point scale:

$$\text{Reward} = \underbrace{R_{\text{CD}}}_{\substack{\text{Spatial} \\ \text{(3pts)}}} + \underbrace{R_{\text{TED}} + R_{\text{HJS}}}_{\substack{\text{Topological} \\ \text{(2pts)}}} \in [0, 5] \tag{6}$$

**Preference Pair Construction.** We generate four candidate skeletons for each input and select preference pairs using our topology-aware reward criterion. Specifically, pairs in which both skeletons reward below a predefined threshold (set to 3) are discarded. When a skeleton outperforms its counterpart by more than 0.5 points, the superior skeleton is chosen as the preferred one. Figure 3 shows some selection cases of our collected preference pairs.

**Reward-guided DPO.** Unlike standard DPO, which leverages only the binary preference order between two responses, our approach incorporates reward differences to exploit richer comparative information. By multiplying the DPO loss with the reward difference, we amplify gradients for pairs with larger disparities, guiding the model to better distinguish between high-fidelity and low-fidelity skeletons and enhance the discriminative gap during optimization. By training on triplets of inputs $x$, high-reward outputs $y_g$, and low-reward outputs $y_b$, the model learns to prioritize generating high-reward samples:

$$\mathcal{L}_{\text{DPO}}(x, y_g, y_b) = -\mathbb{E}_{(x, y_g, y_b) \sim \mathcal{D}} \left[ \log \sigma \left( \beta \log \frac{\pi(y_g|x)}{\pi_{\text{ref}}(y_g|x)} - \beta \log \frac{\pi(y_b|x)}{\pi_{\text{ref}}(y_b|x)} \right) \cdot \left( r^\star(x, y_g) - r^\star(x, y_b) \right) \right] \tag{7}$$

where $\beta$ is a hyperparameter balancing the distance to the reference policy $\pi_{\text{ref}}$, set to 0.3 in our experiments, $\pi$ denotes the policy model being optimized, $r^\star(x, y_g)$ and $r^\star(x, y_b)$ are the rewards assigned by our topology-aware reward function. Furthermore, while high-reward skeletons exhibit better quality, they may still fall short of the perfect ground truth $y_{\text{gt}}$, which obtains the maximum 5-point reward. To address this limitation and ensure the model not only discriminates between good and bad cases but also retains foundational generative capabilities, we introduce an auxiliary SFT loss, where the ground truth $y_{\text{gt}}$ is used to compute the next-token prediction loss. This loss $\mathcal{L}_{\text{SFT}}$ mitigates excessive deviation from the pre-trained knowledge base, ensuring stability during preference alignment. Finally, the total loss for the post-training stage is:

$$\mathcal{L}_{\text{stage2}} = \mathcal{L}_{\text{DPO}}(x, y_g, y_b) + \lambda \mathcal{L}_{\text{SFT}}(x, y_{\text{gt}}) \tag{8}$$

Table 1: **Quantitative comparison on rigging result.** $^*$ denotes models trained on Art-XL2.0 and tested on MR. MagicArticulate and UniRig cannot be trained on the MR dataset as their training script is not provided. ▮ ▮ ▮ indicate the best, second best, and third best performance respectively.

| Method | Dataset | CD-J2J↓ | CD-J2B↓ | CD-B2B↓ | IoU↑ | Prec.↑ | Rec.↑ |
|---|---|---|---|---|---|---|---|
| RigNet | | 7.587% | 6.347% | 6.366% | 21.055% | 21.015% | 33.135% |
| Magic-hier | | 3.435% | 2.757% | 2.393% | 76.364% | 78.121% | 77.567% |
| UniRig | Art-XL2.0 | 3.232% | 2.540% | 2.124% | 75.571% | 77.334% | 77.323% |
| Magic-spatial | | 3.041% | 2.479% | 2.099% | 78.675% | 80.026% | 80.085% |
| Ours | | **2.572%** | **2.030%** | **1.683%** | **82.806%** | **85.254%** | **82.918%** |
| RigNet | | 6.375% | 5.115% | 5.245% | 31.034% | 24.034% | 50.002% |
| Magic-hier$^*$ | | 4.119% | 3.155% | 2.780% | 63.382% | 57.889% | 73.680% |
| UniRig$^*$ | MR | 3.797% | 2.888% | 2.437% | 62.832% | 56.230% | 76.419% |
| Magic-spatial$^*$ | | 3.920% | 3.138% | 2.712% | 63.831% | 57.735% | 75.667% |
| Ours$^*$ | | 3.735% | 2.816% | 2.362% | 66.488% | 62.655% | 75.059% |
| Ours | | **3.203%** | **2.436%** | **2.046%** | **73.108%** | **73.965%** | **76.795%** |

## 3.3 Skinning Weight Prediction

The fundamental limitation of existing skinning methods [33, 46, 71] lies in their static bone selection paradigm—they precompute vertex-bone distances to permanently select the $k$-nearest bones, and assume that the vertex is influenced only by these bones. In contrast, we present a plug-and-play geodesic-aware bone probability prediction module that dynamically identifies the $k$ most probable influencing bones conditioned on implicit geodesic features.

**Geodesic-aware Bone Probability Prediction Module.** The inputs include the vertex positions, the vertex normal, the coordinates of joints, and the vertex-bone geometric distances computed from both raw and laplacian-smoothed meshes. These attributes are processed through a three-layer MLP to predict the influence probabilities of each bone for every vertex, and the $k$ highest-probability bones are selected. To optimize bone selection, we reframe the issue of whether the bone $b_j$ impacts the vertex $v_i$ as a binary classification problem. Here, a label of 1 signifies influence, while a label of 0 indicates no influence. Thus, this module minimizes the discrepancy between the chosen bone and the actual bone using Binary-Cross-Entropy loss: $\mathcal{L}_{\mathrm{S}} = \sum_{i=1}^{N} \left( -\hat{b} \log\left(b\right) - \left(1 - \hat{b}\right) \log\left(1 - b\right) \right)$, where $\hat{b}$ represents the ground truth labels, $b$ denotes the predicted probabilities, and $N$ is the number of vertices. For more details, please refer to the appendix A.1.

**Skinning Weight Prediction Module.** To highlight the plug-and-play nature of our bone probability prediction module, we integrate it with existing skinning methods. We consider the skinning weight matrix as the probability of each vertex binding to each bone. Thus, this module minimizes the discrepancy between the predicted skinning weights distribution and the actual distribution using Kullback-Leibler divergence loss: $\mathcal{L}_{\mathrm{D}} = \sum_{i=1}^{N} \sum_{j=1}^{B} w_{ij} \left( \log \frac{w_{ij}}{\hat{w}_{ij}} \right)$, where $\hat{w}_{ij}$ is the ground truth, $w_{ij}$ is the predicted skinning weights, and $B$ is the number of bones. Finally, the total loss for the skinning weight prediction stage is: $\mathcal{L}_{\mathrm{stage3}} = \mathcal{L}_{\mathrm{S}} + \mathcal{L}_{\mathrm{D}}$.

# 4 Experiments

## 4.1 Implementation Details

**Dataset.** We evaluated our model on Articulation-XL2.0 (Art-XL2.0) [38] and ModelsResource (MR) [33]. Art-XL2.0 provides 46k samples for training and 2k samples for testing, while MR dataset contains 2.1k training samples and 540 testing samples.

**Data augmentation.** To enhance model robustness and generalization, we applied a comprehensive data augmentation strategy during the rigging pre-training stage. This includes the following components: 1) random mesh translations within $[-0.3, 0.3]$, 2) random axial rotations, 3) non-uniform scaling along each axis to introduce variations in proportions, and 4) bone perturbation, where a randomly selected bone is rotated by an angle sampled from a gaussian distribution $N\left(0, 25°\right)$.

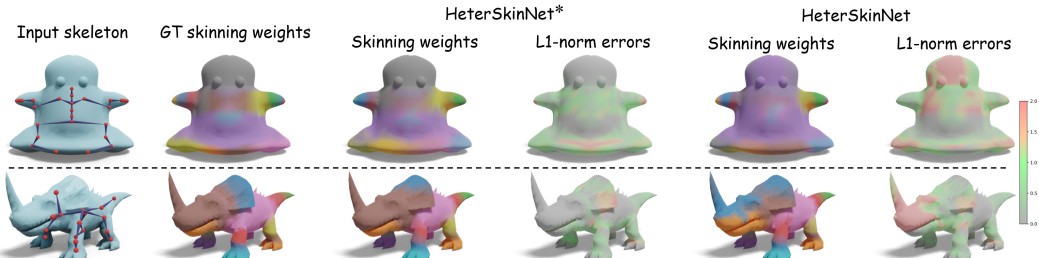

Figure 5: **Qualitative comparison of skinning result on Art-XL2.0 (top) and MR (bottom).** Models marked with * were trained using our geodesic-aware bone probability prediction module, which effectively mitigates the L1-norm error and enhances skinning performance. Additional results are provided in the appendix B.2.

Table 2: **Quantitative comparison on skinning result.** Models marked with * were trained using our geodesic-aware bone probability prediction module. Results for the MR dataset are provided in the appendix B.2.

| Method | Dataset | Prec.↑ | Rec.↑ | avg L1↓ | avg Dist↓ |
|---|---|---|---|---|---|
| RigNet | | 87.47% | 56.04% | 0.52 | 0.0090 |
| RigNet* | | **87.67%** | **59.98%** | **0.44** | **0.0079** |
| NeuroSkinning | Art-XL2.0 | 87.05% | 55.51% | 0.52 | 0.0089 |
| NeuroSkinning* | | **88.00%** | **61.31%** | **0.45** | **0.0080** |
| HeterSkinNet | | 88.16% | 60.18% | 0.43 | 0.0079 |
| HeterSkinNet* | | **89.38%** | **61.13%** | **0.42** | **0.0075** |

**Training details.** The rigging pre-training stage uses a batch size of 192, lasting 2 days on the Art-XL2.0 dataset and 10 hours on the MR dataset, while the post-training stage runs 5 epochs on 14k curated preference pairs. For the skinning weight prediction stage, we set $k = 6$ follow the baseline. Training uses a batch size of 80, lasting 1 day for the Art-XL2.0 dataset and 10 hours for the MR dataset. See the appendix A.2 for more details.

## 4.2 Metrics and Baselines

**Metrics.** Consistent with RigNet [33], we evaluate rigging results using CD-J2J (Chamfer Distance between Joints), CD-J2B (Chamfer Distance between Joints and Bones), CD-B2B (Chamfer Distance between Bones), IoU (Intersection over Union), Precision, and Recall. For skinning results, we adopt Precision, Recall, L1-norm error, and distance error to comprehensively assess bone identification accuracy, skinning weight precision, and deformation quality.

**Baselines.** For rigging results, we compare our approach against state-of-the-art approaches, including RigNet [33], MagicArticulate [38], and UniRig [39]. MagicArticulate is evaluated using both its proposed hierarchical (Magic-hier) and spatial (Magic-spatial) sequence orders. Since RigAnything [40] does not share its code, we cannot compare to it. For skinning results, we integrate our geodesic-aware bone probability prediction module with three top-$k$-based skinning methods—RigNet [33], NeuroSkinning [71], and HeterSkinNet [46]—to demonstrate its compatibility and effectiveness. All vertex-to-bone geodesic distance computations adhere to the HollowDist proposed in the HeterSkinNet, with GPU-accelerated 256-resolution voxelization [72].

## 4.3 Comparison

**Rigging Comparison.** As quantitatively shown in 1, our method achieves comprehensive improvements across all metrics on both datasets. For skeleton location accuracy, we outperform Magic-spatial by +4.2% IoU, +5.2% Precision, and +2.9% Recall on Art-XL2.0, while reducing 15% by CD-J2J, 18% by CD-J2B, and 20% by CD-B2B. The improvements over RigNet are even more pronounced. In terms of topological integrity, as qualitatively shown in Figure 4, MagicArticulate's representation fails to capture the inherent topological relationships within skeletal structures, often

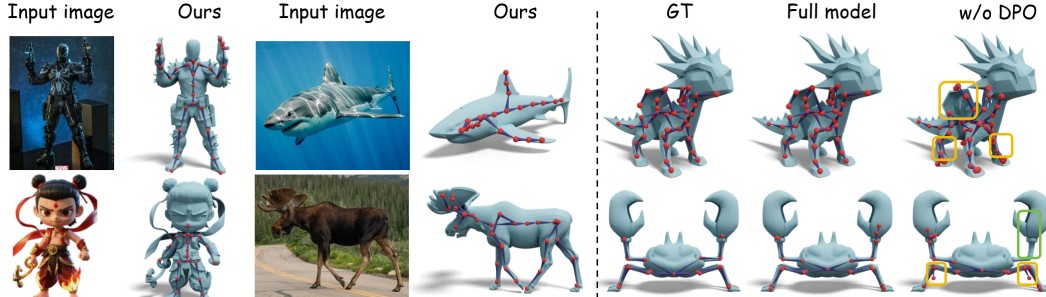

| Input image | Ours | Input image | Ours | GT | Full model | w/o DPO |

Figure 6: **(Left) Rigging results on mesh from in-the-wild images.** We use off-the-shelf image-to-3D model Hunyuan3D 2.5 [3] to generated meshes from input images. **(Right) Ablation study on DPO.** The proposed reward-guided DPO learns human preferences to produce skeletons that better align with artistic aesthetics.

Table 4: **Ablation study on DPO**. Results for the MR dataset are provided in the appendix B.3.

| Method | Dataset | CD-J2J↓ | CD-J2B↓ | CD-B2B↓ | IoU↑ | Prec.↑ | Rec.↑ |
|--------|---------|---------|---------|---------|------|--------|-------|
| w/o DPO | Art-XL2.0 | 2.695% | 2.109% | 1.750% | 82.183% | 84.234% | 82.734% |
| Ours | | **2.572%** | **2.030%** | **1.683%** | **82.806%** | **85.254%** | **82.918%** |

resulting in unconnected joints and discontinuous skeletons (highlighted in orange boxes). UniRig struggles with insufficient spatial continuity between sequentially generated bone chains. Specifically, the large gap between the terminal joint of a completed chain and the expected starting position of the next chain creates initialization barriers, leading to incomplete skeletal structures with missing bone chains (highlighted in green boxes), particularly for characters with tails or wings. MagicArticulate also suffers from similar shortcomings. RigNet, on the other hand, frequently generates an excessive number of joints. In contrast, our method consistently produces accurate, well-structured, and coherent skeletons that closely align with the shapes across diverse categories.

**Skinning Comparison.** Table 2 quantitatively compares baseline performance with and without incorporating the proposed geodesic-aware bone probability prediction module. The results demonstrate the effectiveness of our method in accurately identifying influential bones, reducing L1-norm errors, and preventing incorrect deformations during motion. Furthermore, Figure 5 presents a qualitative comparison of the per-vertex L1-norm errors and predicted skinning weights. The method with the module shows its capability to produce more reliable skinning weights that closely match the ground truth and improve animation fidelity.

### 4.4 Ablation Studies

To validate the effectiveness of the key components of our method, we conducted a series of ablation studies targeting: 1) the proposed connectivity-preserving tokenization strategy, 2) the data augmentation strategy, and 3) the reward-guided DPO post-training strategy.

**Effectiveness of the tokenization strategy.** Table 3 compares the average token sequence length between our method against baselines. Compared to MagicArticulate, our method reduces sequence length by 26% by eliminating redundant parent joint encoding. For UniRig, despite similar compression efficiency, it ignores skeletal topology and relies on manually designed heuristic rules to determine the connections between different bone chains, limiting its generalization and resulting in suboptimal topology quality. In contrast, our method explicitly encodes skeletal topology and connectivity, ensuring better generalization.

Table 3: **Comparison of different tokenization strategies.**

| Method | Dataset | avg Tokens↓ |
|--------|---------|-------------|
| MagicArticulate | | 201.00 |
| UniRig | Art-XL2.0 | **140.26** |
| Ours | | 142.01 |

**Effectiveness of the DPO post-training.** We compared the post-trained model with the pre-trained model to assess the impact of reward-guided DPO. Table 4 quantitatively demonstrates that DPO-enhanced results have achieve higher skeleton location accuracy. Additionally, the right side of Figure 6 qualitatively highlights the topological connectivity improvements introduced by our

topology-aware reward function. Models without DPO often suffer from missing skeletal elements, such as the crab's right claw (highlighted in green boxes), or exhibit disorganized structures, such as wings, feet, and the crab's legs (highlighted in orange boxes). In contrast, the full model produces more complete and structured results, aligning well with artistic preferences.

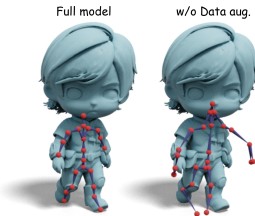

Figure 7: **Ablation study on data augmentation.**

**Effectiveness of the data augmentation.** Figure 7 compares results with and without our data augmentation strategy on a character in a walking pose not present in the dataset. Our full model generates skeletons that are better aligned with the character's shape, whereas the model trained without augmentation struggles. Furthermore, as shown on the left side of Figure 6, our model effectively handles cartoon characters and in-the-wild data, highlighting the augmentation's role in improving generalization to unseen poses and data types.

## 5 Conclusion

We present Auto-Connect, a method that transforms static 3D meshes into animation-ready assets. By integrating connectivity-preserving tokenization, reward-guided fine-tuning, and geodesic-aware bone selection, our approach achieves exceptional rigging and skinning performance across diverse categories of 3D meshes. We believe Auto-Connect holds great potential to revolutionize 3D content creation, offering a more efficient solution for digital artists by streamlining animation workflows. Limitations and future work are discussed in appendix C.

## Acknowledgements

This work was supported in part by the National Natural Science Foundation of China under Grant 62202174, in part by the GJYC program of Guangzhou under Grant 2024D01J0081, and in part by the ZJ program of Guangdong under Grant 2023QN10X455, and in part by the Fundamental Research Funds for the Central Universities under Grant 2025ZYGXZR053.

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

# Appendix

In this appendix, we provide additional content to complement the main manuscript, including:

- Further details of Auto-Connect (Section A).
- Additional experimental results on rigging and skinning (Section B).
- A discussion of the limitations of our work and future works (Section C).

## A    More Details of Auto-Connect

### A.1    More Details of Geodesic-aware Bone Probability Prediction Module

This module combining heterogeneous features, including vertex positions $p_v$, vertex normal $n_v$, the coordinate of the bone start and end joints $\left(p_{b_j^s}, p_{b_j^e}\right)$, and the vertex-bone geometric distances computed from both raw and laplacian-smoothed meshes. These features are processed by an three-layer MLP to predict bone influence probabilities:

$$\tilde{b} = \text{top\_k}\Big(\text{MLP}\big(p_v, n_v, p_{b_j^s}, p_{b_j^e}, d_v^{b_j}, d_{v_{l5}}^{b_j}, d_{v_{l10}}^{b_j} \,\big|\, b_j \in \mathcal{B}\big)\Big) \tag{9}$$

where $d_{v_{l5}}^{b_j}$ and $d_{v_{l10}}^{b_j}$ denote distances after performing 5/10 laplacian smoothing iterations, respectively. The resulting selected bone set $\tilde{b}$ for each vertex $v$ is used as the input to the skinning weight prediction module to compute the final skinning weights.

### A.2    More Training Details

The rigging pre-training stage is conducted with a global batch size of 192, lasting 2 days on the Art-XL2.0 dataset and 10 hours on the MR dataset. We use the Adam optimizer with a base learning rate of $5 \times 10^{-5}$, a weight decay of 0.001, and a linear warmup for the first 1,000 steps. For the DPO post-training stage, the optimizer remains unchanged, but the learning rate is reduced to $1 \times 10^{-6}$, and the coefficient for $\mathcal{L}_{\text{SFT}}$ is set to $\lambda = 1$. This stage performs 5 epochs on 14k curated preference pairs. The RigFormer model consists of 24 layers with a hidden dimension of 1024, and each transformer block incorporates a 16-head multi-head self-attention mechanism.

For skinning weight prediction, following the baseline, we set $k = 6$ nearest bones and prune non-influential joints during ground truth construction. Training is conducted on $8 \times$ H20 GPUs with a global batch size of 80, lasting 1 day for the Art-XL2.0 dataset and 10 hours for the MR dataset. The geodesic-aware bone probability prediction module is implemented as a three-layer MLP with a hidden dimension of 256 and ReLU activation.

For the baselines MagicArticulate [38] and UniRig [39], we utilize their publicly available pre-trained weights on the Art-XL2.0 dataset for comparison since their training scripts are not provided. All other baselines [33, 46, 71] are retrained on the same datasets for a fair comparison.

### A.3    Animation Details

Auto-Connect provides an automated animation pipeline. The resulting animation-ready assets can be exported in standard formats such as FBX and GLB. These assets are directly compatible with popular animation software like Blender [73] and Autodesk Maya [74], enabling digital artists to edit and refine them. Animation videos are included in the supplementary materials.

## B    Additional Experimental Results

### B.1    More Rigging Result

We provide additional qualitative rigging results on both Articulation-XL and ModelsResource datasets. As illustrated in Fig. 11, our method consistently generates high-quality skeletons, even

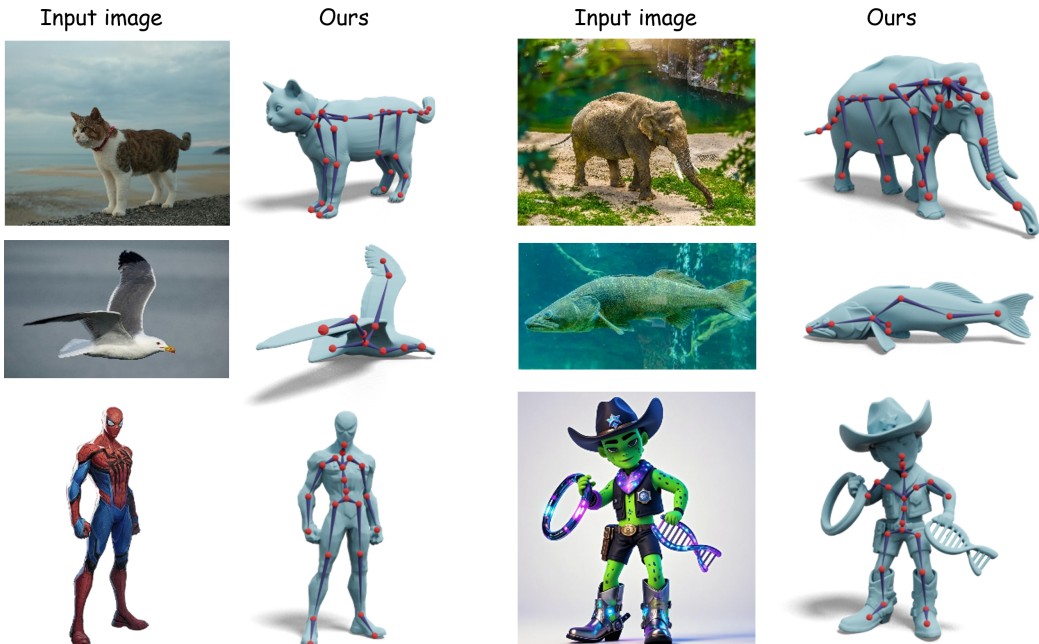

Figure 8: **More rigging results on mesh from in-the-wild images.** We use off-the-shelf image-to-3D model Hunyuan3D 2.5 [3] to generated mesh from the input images. The rigging results demonstrate that our model has strong generalization to unseen data, achieving artist-approved skeleton quality and transforming static 3D meshes into animation-ready assets.

Table 5: **Quantitative comparison of skinning result on ModelsResource dataset.** Models marked with * were trained using our geodesic-aware bone probability prediction module.

| Method | Dataset | Prec.↑ | Rec.↑ | avg L1↓ | avg Dist↓ |
|---|---|---|---|---|---|
| RigNet | | 86.03% | 79.03% | 0.36 | 0.0058 |
| RigNet* | | **87.85%** | **80.11%** | **0.33** | **0.0049** |
| NeuroSkinning | MR | 86.24% | 78.31% | 0.36 | 0.0057 |
| NeuroSkinning* | | **88.03%** | **79.27%** | **0.33** | **0.0049** |
| HeterSkinNet | | 87.31% | 78.99% | 0.34 | 0.0052 |
| HeterSkinNet* | | **89.09%** | **79.45%** | **0.28** | **0.0045** |

for complex cases. In contrast, the baseline methods produce suboptimal results that are not directly suitable for animation pipelines. Fig. 8 showcases more rigging results on AI-generated 3D data, further validating the robustness and effectiveness of our approach.

## B.2 More Skinning Results

Table 5 presents a quantitative comparison of baseline performance with and without incorporating this module on the ModelsResource dataset. The results demonstrate substantial improvements across all evaluation metrics, including higher precision, higher recall, lower L1-norm error, and reduced distance error. This highlights the effectiveness of our approach in enhancing skinning accuracy. In addition, Fig. 12 illustrates qualitative comparisons, where the integration of our module enables the baseline to produce smoother and more realistic skin deformations, particularly in regions with complex geometries. These results further highlight the module's ability to accurately identify influence bones, leading to more precise predictions of skinning weights with minimal L1-norm error.

## B.3 More Ablation Studies

**More Ablation study on DPO post-training.** Table 6 presents a quantitative comparison of model performance with and without the proposed DPO post-training on the ModelsResource

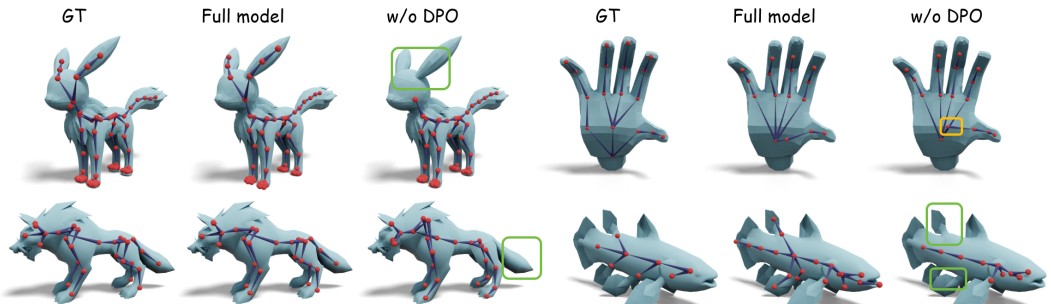

Figure 9: **More qualitative ablation study on DPO post-training.** Models trained without DPO often suffer from missing details—such as ears, tails, or fins (highlighted in green boxes)—or generate structural artifacts like crossing topologies (highlighted in orange boxes). In contrast, our full model effectively alleviates these issues, producing well-defined skeletons that better align with the artistic aesthetics expected by creators.

Table 6: **Quantitative ablation study of DPO post-training on ModelsResource dataset.**

| Method | Dataset | CD-J2J↓ | CD-J2B↓ | CD-B2B↓ | IoU↑ | Prec.↑ | Rec.↑ |
|--------|---------|---------|---------|---------|------|--------|-------|
| w/o DPO | MR | 3.426% | 2.576% | 2.164% | 72.213% | 73.700% | 72.822% |
| Ours | | **3.203%** | **2.436%** | **2.046%** | **73.108%** | **73.965%** | **76.795%** |

dataset. The results clearly demonstrate that our reward-guided DPO post-training substantially improves the model's accuracy in skeleton localization, highlighting the effectiveness of the proposed approach. In addition, Fig. 9 showcases additional qualitative evidence from ablation studies. Our full model produces more realistic topology connections and generates more complete skeletons, closely resembling the ground truth.

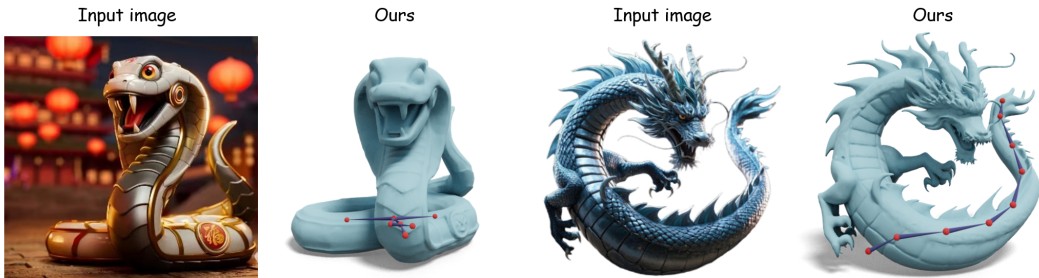

Figure 10: **Some failure cases.** Our method struggles with snake-like data, the input meshs are generated using Hunyuan3D 2.5 [3].

## C Limitations and Future Work

While our method achieves significant improvements in skeleton location accuracy, topological consistency, and skinning quality compared to prior approaches, it still has some limitations. As shown in Fig. 10, the main drawback is the inability to generalize well to snake-like data, due to the lack of such samples in the training dataset. Future work could address this by expanding the dataset with more snake-like examples or by developing more robust data augmentation techniques. Another promising direction is leveraging multimodal input, such as text, image, and video, to allow user-friendly editing of rigging results.

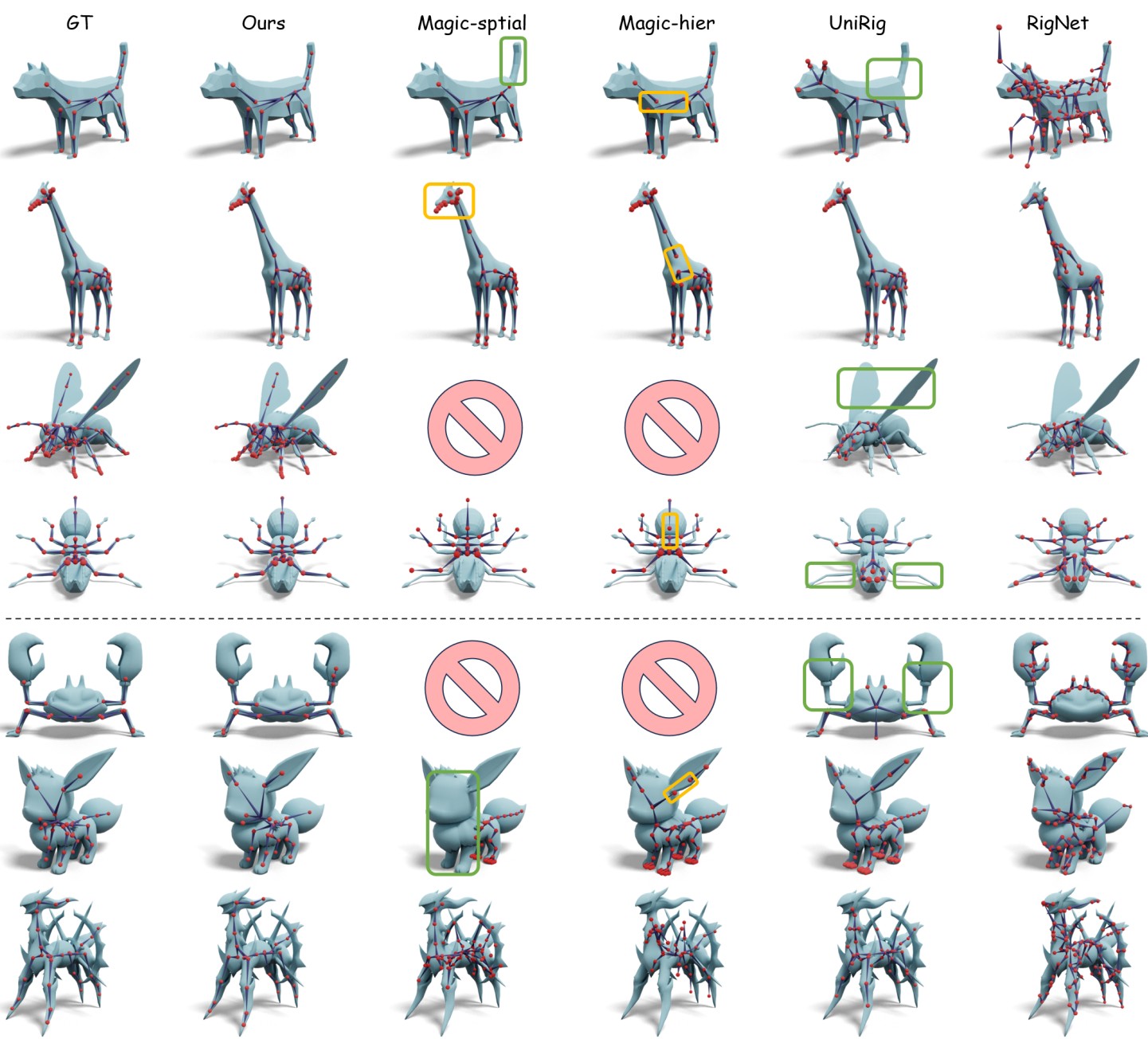

Figure 11: **More qualitative comparison of rigging results on Art-XL2.0 (top) and ModelsResource (bottom).** MagicArticulate often generates discontinuous skeletons (highlighted in green boxes) or even fails entirely. UniRig tends to predict the termination token prematurely, leading to incomplete skeletal structures with missing bone chains (highlighted in orange boxes), and RigNet frequently produces overly dense joints, resulting in disorganized skeleton topologies. In contrast, our method reliably generates coherent, accurate, and well-structured skeletons that closely conform to the shapes.

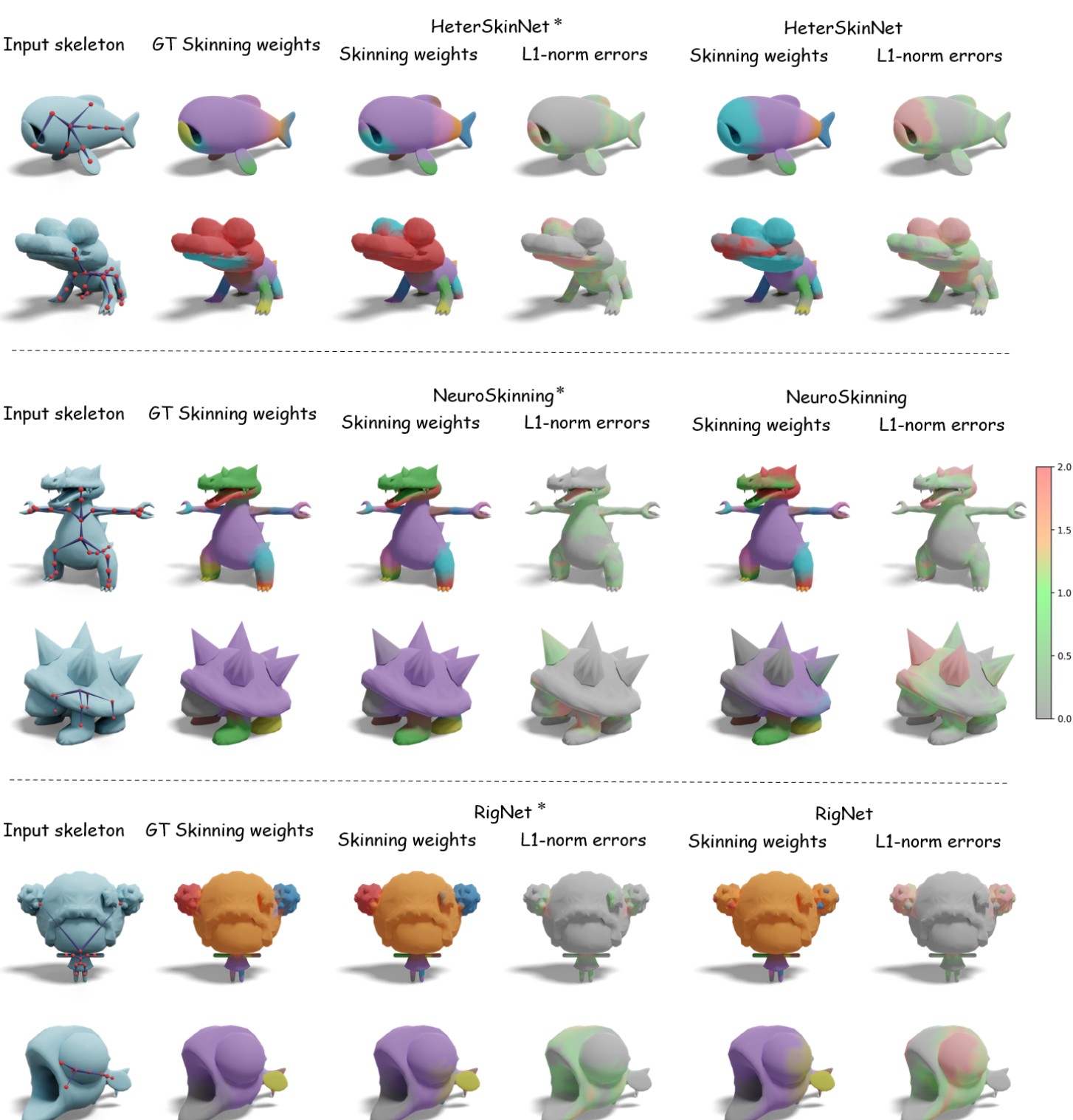

Figure 12: **More qualitative comparison of skinning result on ModelsResource dataset.** Models marked with * were trained using our geodesic-aware bone probability prediction module. By incorporating this module, the baseline method achieves lower L1-norm errors and more precise skinning weights, resulting in more accurate deformations during animation.

