# OpenReview forum: "Auto-Connect: Connectivity-Preserving RigFormer with  Direct Preference Optimization"
_NeurIPS.cc/2025/Conference — NeurIPS 2025 poster_

### Official Review · Reviewer_LNzV · 2025-06-18

**Clarity:** 4
**Significance:** 3
**Originality:** 2
**Rating:** 4
**Confidence:** 4

**Summary:**

The paper aims to solve the problem of 3D mesh rigging. This method primarily consists of 3 key components: 1) a hierarchical skeletal representation, 2) DPO-based post-training to ensure better rigging refinement, and 3) a novel skinning weight prediction module. The paper conducts extensive experiments, achieving superior results over existing methods both qualitatively and quantitatively, demonstrating the effectiveness of this approach.

**Questions:**

Overall, I recommend that the authors:
1. Explain why the results in this paper are inconsistent with the original MagicArticulate paper.
2. Provide a more thorough explanation of the similarities and differences between their token representation and existing works, along with an analysis of its merits and limitations.
3. Supplement additional ablation results to thoroughly validate the effectiveness of the proposed modules.

As for the DPO-based post-training, the current framework, after sampling shapes, may not fully capture the semantic information of the mesh, specifically, whether certain parts should be rigged with bones. I wonder if incorporating richer semantic information could enhance the quality of skeleton prediction.

**Ethical Concerns:**

["NO or VERY MINOR ethics concerns only"]

**Final Justification:**

Thank the authors for their responses, which have effectively addressed my concerns. However, as the results of the ablation study show, data augmentation has brought a significant improvement to the proposed method. Considering the effect brought in post-training, I increase the rating to Borderline Accept.

**Limitations:**

yes

**Quality:**

3

**Strengths And Weaknesses:**

Strengths:
1. This work introduces DPO-based post-training for refinement, which is intuitive and effective.
2. The results of this paper are excellent, outperforming existing works both qualitatively and quantitatively.
3. The paper is well-written, easy to understand and follow.

Weaknesses:
1. In Table 1, the results for Magic-spatial and Magic-hier are significantly worse than the results reported in their original papers. Specifically, Magic-spatial yields better results than the proposed method. Please confirm it.
2. The ablation studies seem insufficient. For example, comparisons with baseline tokenization methods (e.g., UniRig's), and validation on the data augmentation. The authors should provide complete quantitative comparisons for these ablations.
3. Table 4 indicates that DPO provides only limited quantitative gains. Does pre-training alone achieve results far superior to the current SOTA? The main improvement of the token is the addition of  \< E2 \>  compared to other BFS skeletons. The rigging results of hands in Fig. 4 and Fig. 9 seem contradictory.
4. Performance on in-the-wild data is not good. For instance, in the first row of Table 6 (left), there should not be bones on the guns. In the second row, the hand bones are incorrect. I intuitively feel that the proposed DPO encourages skeleton expansion but overlooks semantic information.

---

> ### Author Rebuttal · Authors · 2025-07-29
>
> We thank you for your constructive comments and appreciation, such as 'the DPO-based post-training refinement is intuitive and effective', 'the paper is well-written', and 'the results of this paper are excellent'. We hope the following responses address your concerns, and we’re happy to provide further clarification if needed.
>
>
> > **Weaknesses 1 & Questions 1: In Table 1, the results for Magic-spatial and Magic-hier are significantly worse than the results reported in their original papers. Specifically, Magic-spatial yields better results than the proposed method. Please confirm it. Explain why the results in this paper are inconsistent with the original MagicArticulate paper.**
>
> **Answer:**  This is due to different versions of the Articulation-XL dataset. The original MagicArticulate paper used Articulation-XL 1.0 (contains 33k samples, with 31.4k for training and 1.6k for testing), whereas our results are evaluated on the more recent and larger Articulation-XL 2.0 (contains 48k samples, with 46k for training and 2k for testing). The results reported in Table 1 of the main paper were obtained using the official MagicArticulate code and checkpoint on the Articulation-XL 2.0 test set. The performance of MagicArticulate on Articulation-XL 2.0 is also documented in the README of their repo. We ensured a fair comparison by evaluating all baselines and our method on the same settings on the Articulation-XL 2.0 dataset.
>
>
>
> > **Weaknesses 2 & Questions 3: The ablation studies seem insufficient. For example, comparisons with baseline tokenization methods (e.g., UniRig's), and validation on the data augmentation. The authors should provide complete quantitative comparisons for these ablations. Supplement additional ablation results to thoroughly validate the effectiveness of the proposed modules.**
>
> **Answer:** In fact, our experiments include a comparison of the proposed tokenization strategy. The main difference between our pre-training stage and those of MagicArticulate and UniRig lies in the tokenization approach, while the model architecture remains a standard transformer. This means that comparing our w/o DPO (stage 1) rigging results directly against MagicArticulate and UniRig effectively serves as a comparison of the tokenization strategy. The results without the DPO stage are reported in Table 4 of the main paper, and baseline results is reported in Table 1 of the main paper. For clarity, we summarize them in Table 1 below:
>
> **Table 1: Comparison on different tokenization methods**
> | Method        | Dataset    | CD-J2J↓ | CD-J2B↓ | CD-B2B↓ | IoU↑    | Prec.↑  | Rec.↑   |
> |:--------|:----------|:-------|:-------|:-------|:------|:------|:-------|
> | Magic-hier   | Art-XL2.0 | 3.435% | 2.757% | 2.393% | 76.364% | 78.121% | 77.567% |
> | UniRig       | Art-XL2.0 | 3.232% | 2.540% | 2.124% | 75.571% | 77.334% | 77.323% |
> | Magic-spatial| Art-XL2.0 | 3.041% | 2.479% | 2.099% | 78.675% | 80.026% | 80.085% |
> | **Ours stage 1** | **Art-XL2.0** | **2.695%** | **2.109%** | **1.750%** | **82.183%** | **84.234%** | **82.734%** |
>
>
> Additionally, we provide a quantitative ablation on the effect of data augmentation in Table 2 below:
> **Table 2: Ablation on the effect of data augmentation**
> | Method   | Dataset    | CD-J2J↓ | CD-J2B↓ | CD-B2B↓ | IoU↑   | Prec.↑ | Rec.↑   |
> |:--------|:----------|:-------|:-------|:-------|:------|:------|:-------|
> | Ours w/o Data aug. | Art-XL2.0  | 2.834%  | 2.351%  | 1.827%  | 81.017%| 82.835%| 81.393% |
> | **Ours Full model**    | **Art-XL2.0**  | **2.572%** | **2.030%** | **1.683%** | **82.806%** | **85.254%** | **82.918%** |
>
>
>
> > **Weaknesses 3: Table 4 indicates that DPO provides only limited quantitative gains. Does pre-training alone achieve results far superior to the current SOTA?**
>
> **Answer:** Yes, our pre-trained model already significantly outperforms prior SOTA methods, as shown in Table 1 in the answer for Weaknesses 2 & Questions 3. While DPO brings limited improvements in quantitative metrics, which primarily reflect joint localization accuracy, its main contribution lies in enhancing connectivity fidelity, as shown in Figure 6 right. This leads to more coherent and topologically accurate skeletons that better match the expectations of designers and technical artists—an improvement not fully captured by standard metrics.
>
>
> > **Weaknesses 4 & Questions 2: The main improvement of the token is the addition of <E2\> compared to other BFS skeletons. Provide a more thorough explanation of the similarities and differences between their token representation and existing works, along with an analysis of its merits and limitations.**
>
> **Answer:** Below, we detail the tokenization strategies used in baselines and point out their limitations, then highlight the advantages of our approach. Table 3 summarizes the key differences.
>
> **Similarities**
> All tokenization strategies aim to serialize the skeleton into a sequence for Transformer-based modeling.
>
> **Differences**
> The key differences lie in the sequence ordering strategy and the reliance on heuristic post-processing to recover connectivity. These are summarized in Table 3 below:
> **Table 3: Summary of differences in tokenization strategies.** T: number of bones; S: number of branches; M: number of special tokens (typically close to S$\times$4).
> | Method   | Number of tokens  | Order  | Post-Processing Required |
> |:--------|:----------|:-------|:-------|
> | MagicArticulate    | 6T | Spatial/Hierarchical | Yes |
> | Unrig | 3T + S$\times$4| DFS + <branch token\>| Yes  |
> | Ours  | 3T + M | BFS + <special token\> | No |
>
> **Limitations of Baselines**
> (1) MagicArticulate employs a bone-wise tokenization strategy with two sequence ordering variants:
> * Spatial sequence ordering: Bones are sorted based on the spatial location of their joints, first by the lower joint, then by the higher one.
> * Hierarchical sequence ordering: Bones are ordered based on the skeleton hierarchy. Starting from the root joint, bones connected to it are ordered by ascending child joint index. For subsequent layers, bones are grouped by their immediate parent and sorted within each group by child joint index; the groups themselves are processed in ascending parent joint order.
>
> Both variants requires heuristic post-processing to determine connections between bones. Specifically, it merges bones whose endpoints lie within a predefined threshold. If bones fall outside this threshold, the connections fail, leading to disconnected skeletons. Additionally, the method re-encode parent joints multiple times, leading to longer sequences and slower inference times.
>
> (2) UniRig decomposes the skeleton into linear bone chains via Depth-First Search (DFS), each prefixed with a <branch token\>. However, it similarly relies on heuristic post-processing to connect these chains. Moreover, because the end of one chain and the start of another are often spatially and sequentially distant, the method frequently fails to initiate subsequent chains, resulting in an incomplete or fragmented skeleton.
>
> **Merits of Our Method**
> The main improvement is not merely the addition of <E2\> compared to other BFS skeletons, but by introducing special tokens to explicitly mark endpoints for each joint’s children and each hierarchical layer. This allows the transformer to directly encode both the skeleton topology and parent-child relationships within its representation. Consequently, the model gains stronger topology awareness, eliminates the need for heuristic post-processing, and can automatically infer joint connectivity, leading to more coherent and complete skeleton structures.
>
>
>
>
>
>
>
> > **Weaknesses 5: The rigging results of hands in Fig. 4 and Fig. 9 seem contradictory.**
>
> **Answer:** The rigging results in Figure 4 and Figure 9 differ because the two hand meshes are structurally distinct, and their ground-truth skeletons are also different.
>
>
>
>
>
> > **Weaknesses 6: Performance on in-the-wild data is not good. For instance, in the first row of Figure 6 (left), there should not be bones on the guns. In the second row, the hand bones are incorrect. I intuitively feel that the proposed DPO encourages skeleton expansion but overlooks semantic information.**
>
> **Answer:** Our DPO stage does not encourage skeleton expansion but rather promotes correct joint connectivity, as evidenced by both quantitative (Table 4 in the main paper) and qualitative (Figure 6 right in the main paper) improvements.
> The issues in Figure 6 are unrelated to DPO. They arise because our training data lacks meshes with character-object interactions (e.g., holding weapons), making it difficult for the model to distinguish between body parts and accessories. Additionally, these meshes are AI-generated and therefore exhibit inherent domain gaps compared to the artist-designed meshes in our training set. This is a common limitation across all existing methods, including baselines. Addressing it will require more diverse training data with such interactions and semantic annotations.
>
>
> > **Questions 4: As for the DPO-based post-training, the current framework, after sampling shapes, may not fully capture the semantic information of the mesh, specifically, whether certain parts should be rigged with bones. I wonder if incorporating richer semantic information could enhance the quality of skeleton prediction.**
>
> **Answer:** Thank you for the insightful observation. We agree that incorporating semantic information—such as identifying which regions should or should not be rigged—could intuitively improve skeleton prediction, especially in scenarios involving human-object interactions. While our current framework relies primarily on geometric and topological cues, extending it with richer semantic understanding is a valuable and promising direction for future work.

---

> > ### Comment · Reviewer_LNzV · 2025-08-05
> >
> > Thank the authors for their responses, which have effectively addressed my concerns. However, as the results of the ablation study show, data augmentation has brought a significant improvement to the proposed method. Considering the effect brought in post-training, I increase the rating to Borderline Accept.

---

> > > ### Author Response · Authors · 2025-08-08
> > >
> > > Dear Reviewer LNzV,
> > >
> > > Thank you for your thoughtful review and for increasing your rating. We are pleased that our responses have effectively addressed your concerns.
> > >
> > > We will incorporate your excellent suggestions in the revised manuscript. Specifically, we will add a quantitative ablation study on data augmentation and offer a more detailed explanation of the similarities and differences between the baselines and our method to further enhance the paper’s clarity.
> > >
> > > Thank you again for your constructive feedback.

---

### Official Review · Reviewer_8uXR · 2025-06-24

**Clarity:** 3
**Significance:** 2
**Originality:** 2
**Rating:** 4
**Confidence:** 4

**Summary:**

This paper introduces a novel method for automatic rigging of 3D meshes. The authors claim three main contributions: (a) a connectivity-preserving tokenization scheme that converts a skeleton tree into a token sequence by performing BFS traversal and appending special tokens; (b) a post-training phase that aligns model outputs with topology-aware rewards, complementing the next-token prediction strategy used in pretraining; and (c) a geodesic distance-based MLP module for bone prediction in skinning weight estimation.

**Questions:**

1. **What is the relationship between Bone Probability Prediction and Skinning Weight Prediction?** Are you first predicting a set of bones for each vertex, and then predicting the skinning weights for these vertex-bone pairs? Or are you directly predicting skinning weights over all bone-vertex pairs? Additionally, what is the relationship between the Binary Cross-Entropy loss (line 229) and the Kullback-Leibler Divergence loss (line 236)? Are they computed over the same set of variables (i.e., the skinning weights $w_{ij}$)?

2. **Line 192:** Given a set of predicted joints and a set of ground-truth joints, how do you determine which joints are considered *shared joints*?

**Ethical Concerns:**

["NO or VERY MINOR ethics concerns only"]

**Limitations:**

yes

**Paper Formatting Concerns:**

no.

**Quality:**

3

**Strengths And Weaknesses:**

**Strengths**

1. The paper is clear and easy to follow.
2. The proposed techniques are well-motivated and technically sound.
3. The method demonstrates superior performance over baselines on two datasets.
4. The paper shows that the proposed skinning strategy can be coupled with various existing methods and provides performance gains.
5. The method performs well on in-the-wild meshes (e.g., those generated by AI models).

---

**Weaknesses**

1. The current ablation of the tokenization strategy is weak, despite it being one of the main contributions. It only compares the average token length. It is necessary to explicitly describe the tokenization strategies used in baseline methods and compare model performance using different strategies both qualitatively and quantitatively.

2. Figure 7 does not sufficiently demonstrate the benefits of data augmentation, as it only showcases results with data augmentation and lacks results without it. Including quantitative ablation studies on data augmentation would strengthen the paper.

3. To make the paper self-contained, a brief introduction to how DPO works should be included. For example, explain how the reference policy is selected and the relationship between the reference policy and the optimized policy.

4. An explanation of how skinning weights are visualized in Figure 5 would be helpful for readers.

5. In the method section, many abbreviations (e.g., DPO, SFT, CD-J2J, CD-J2B, CD-B2B) should be defined upon their first appearance.

---

> ### Author Rebuttal · Authors · 2025-07-28
>
> We thank you for your constructive comments and appreciation, such as 'well-motivated and technically sound', 'superior performance', and 'the proposed skinning strategy provides performance gains'. We hope the following responses address your concerns, and we’re happy to provide further clarification if needed.
>
>
> > **Weaknesses 1: The current ablation of the tokenization strategy is weak, despite it being one of the main contributions. It only compares the average token length. It is necessary to explicitly describe the tokenization strategies used in baseline methods and compare model performance using different strategies both qualitatively and quantitatively.**
>
> **Answer:** Below, we first describe the tokenization strategies used in existing baselines and point out their limitations, then clarify how our comparative experiments already provide an effective evaluation of different tokenization strategies.
>
> **Tokenization strategies description**
> (1) MagicArticulate employs a bone-wise tokenization strategy with two sequence ordering variants:
> * Spatial sequence ordering: Bones are sorted based on the spatial location of their joints, first by the lower joint, then by the higher one.
> * Hierarchical sequence ordering: Bones are ordered based on the skeleton hierarchy. Starting from the root joint, bones connected to it are ordered by ascending child joint index. For subsequent layers, bones are grouped by their immediate parent and sorted within each group by child joint index, the groups themselves are processed in ascending parent joint order.
>
> Both variants requires heuristic post-processing to determine connections between bones. Specifically, it merges bones whose endpoints lie within a predefined threshold. If bones fall outside this threshold, the connections fail, leading to disconnected skeletons. Additionally, the method re-encode parent joints multiple times leading to longer sequences and slower inference times.
>
> (2) UniRig decomposes the skeleton into linear bone chains via Depth-First Search (DFS), each prefixed with a <branch token\>. However, it similarly relies on heuristic post-processing to connect these chains. Moreover, because the end of one chain and the start of another are often spatially and sequentially distant, the method frequently fails to initiate subsequent chains, resulting in an incomplete or fragmented skeleton.
>
> **Tokenization strategies comparison**
> In fact, our experiments include a comparison of the proposed tokenization strategy. The main difference between our pre-training stage and those of MagicArticulate and UniRig lies in the tokenization approach, while the model architecture remains a standard transformer. This means that comparing our w/o DPO (stage 1) rigging results directly against MagicArticulate and UniRig effectively serves as a comparison of the tokenization strategy. The results without the DPO stage are reported in Table 4 of the main paper, and baseline results is reported in Table 1 of the main paper. For clarity, we summarize them below:
>
> | Method        | Dataset    | CD-J2J↓ | CD-J2B↓ | CD-B2B↓ | IoU↑    | Prec.↑  | Rec.↑   |
> |:--------|:----------|:-------|:-------|:-------|:------|:------|:-------|
> | Magic-hier   | Art-XL2.0 | 3.435% | 2.757% | 2.393% | 76.364% | 78.121% | 77.567% |
> | UniRig       | Art-XL2.0 | 3.232% | 2.540% | 2.124% | 75.571% | 77.334% | 77.323% |
> | Magic-spatial| Art-XL2.0 | 3.041% | 2.479% | 2.099% | 78.675% | 80.026% | 80.085% |
> | **Ours stage 1** | **Art-XL2.0** | **2.695%** | **2.109%** | **1.750%** | **82.183%** | **84.234%** | **82.734%** |
>
>
>
>
> > **Weaknesses 2: Figure 7 does not sufficiently demonstrate the benefits of data augmentation, as it only showcases results with data augmentation and lacks results without it. Including quantitative ablation studies on data augmentation would strengthen the paper.**
>
> **Answer:** In fact, the right side of Figure 7 in the main paper labeled 'w/o Data aug.' refers to the setting without data augmentation. Additionally, we provide a quantitative ablation on the effect of data augmentation in the table below:
>
> | Method   | Dataset    | CD-J2J↓ | CD-J2B↓ | CD-B2B↓ | IoU↑   | Prec.↑ | Rec.↑   |
> |:--------|:----------|:-------|:-------|:-------|:------|:------|:-------|
> | Ours w/o Data aug. | Art-XL2.0  | 2.834%  | 2.351%  | 1.827%  | 81.017%| 82.835%| 81.393% |
> | **Ours Full model**     | **Art-XL2.0**  | **2.572%** | **2.030%** | **1.683%** | **82.806%** | **85.254%** | **82.918%** |
>
>
> > **Weaknesses 3: To make the paper self-contained, a brief introduction to how DPO works should be included. For example, explain how the reference policy is selected and the relationship between the reference policy and the optimized policy.**
>
> **Answer:** Thanks for this comment, we will incorporate this introduction into the revision. Specifically, DPO is a method for post-training using preference data without explicitly learning a reward model. It operates by directly optimizing the policy to align with human preferences through a loss function derived from preference comparisons. The preference pairs consist of different outputs generated from the same input, paired and labeled as "good" and "bad" either manually or based on reward scores. The reference policy, typically the pre-trained model, serves as a fixed baseline to prevent excessive deviation, while the optimized policy is fine-tuned to increase the likelihood of preferred responses. In our case, both the reference and optimized policies are initialized from the same pre-trained model obtained after stage 1. The reference policy is frozen and serves as a stable baseline to prevent excessive deviation, while the optimized policy remains trainable and is further fine-tuned using both DPO and supervised fine-tuning (SFT) losses to generate more plausible and accurate skeletons.
>
>
>
> > **Weaknesses 4: An explanation of how skinning weights are visualized in Figure 5 would be helpful for readers.**
>
> **Answer:** Thanks for this comment, we will update in the revision. The skinning weight visualization follows a three-step process:
> (1) Color assignment: each joint is randomly assigned a unique color from a predefined palette.
> (2) Weighted color blending: vertex colors are computed as a linear combination of joint colors, proportionally weighted by each vertex's skinning weights, donated as $\text{Color}(v_i) = \sum_{j=1}^{J} w_{i,j} \cdot \text{Color}(b_j)$, where $J$ is the number of joints, $w_{i,j}$ is the skinning weight of vertex $v_i$ to joint $b_j$, and $\text{Color}(b_j)$  is the color assigned to joint $b_j$.
> (3) Mesh rendering: the computed vertex colors are applied to the mesh and visualized using smooth shading in Blender.
>
>
> > **Weaknesses 5: In the method section, many abbreviations (e.g., DPO, SFT, CD-J2J, CD-J2B, CD-B2B) should be defined upon their first appearance.**
>
> **Answer:** Thanks for this comment, we will update in the revision.
>
>
>
> > **Questions 1: What is the relationship between Bone Probability Prediction and Skinning Weight Prediction? Are you first predicting a set of bones for each vertex, and then predicting the skinning weights for these vertex-bone pairs? Or are you directly predicting skinning weights over all bone-vertex pairs?**
>
> **Answer:** The Bone Probability Prediction and Skinning Weight Prediction modules operate sequentially. At each step, the Bone Probability Prediction Module first estimates the probability that each bone influences a given vertex. Then, the top-$k$ bones with the highest probabilities are selected. These selected bones are then passed to the Skinning Weight Prediction Module, which predicts the skinning weights between the given vertex and this subset of bones.
>
>
>
> > **Questions 2: Additionally, what is the relationship between the Binary Cross-Entropy loss (line 229) and the Kullback-Leibler Divergence loss (line 236)? Are they computed over the same set of variables (i.e., the skinning weights $w_{ij}$)?**
>
> **Answer:** These two losses are applied to different modules and operate independently. The Binary Cross-Entropy (BCE) loss is applied to the Bone Probability Prediction Module, guiding it to correctly identify the set of bones that potentially influence the given vertex. Specifically, as described in lines 226–229 of the main paper, for each vertex, influencing bones are labeled as 1 and non-influencing bones as 0. We binarize the predicted probabilities by selecting the top-$k$ highest values as 1, with others set to 0, and compute the BCE loss against these ground-truth binary labels.
> The Kullback-Leibler (bones) Divergence loss is used in the Skinning Weight Prediction Module to ensure accurate prediction of the skinning weights between vertices and the given set of bones.
> These two losses are computed on different variables, BCE loss is calculated between the predicted probabilities $b$ and the ground truth labels ${\hat b}$, while the KL loss is computed between the predicted skinning weights $w_{ij}$ and the ground truth skinning weights ${\hat w}_{ij}$.
>
>
>
>
> > **Questions 3: Line 192: Given a set of predicted joints and a set of ground-truth joints, how do you determine which joints are considered shared joints?**
>
> **Answer:** We determine shared joints by traversing both the predicted and ground-truth skeletons using our tokenization order. During this traversal, we assign new names to joints based on their positions in the hierarchy: we perform a BFS traversal of the skeleton, and within each layer, joints are sorted by their height along the z-axis. Joints with matching names after this process are considered shared.

---

> > ### Comment · Reviewer_8uXR · 2025-08-06
> > **thank you**
> >
> > thank you for the clarification.

---

> > > ### Author Response · Authors · 2025-08-08
> > >
> > > Dear Reviewer 8uXR,
> > >
> > > Thank you for your thoughtful review and your positive rating. We are pleased that our responses have effectively addressed your concerns.
> > >
> > > We will incorporate your excellent suggestions in the revised manuscript. Specifically, we will add a quantitative ablation study on data augmentation, provide a brief introduction to DPO, and explain how the skinning weights are visualized to further enhance the paper’s clarity.
> > >
> > > Thank you again for your constructive feedback.

---

### Official Review · Reviewer_c5dz · 2025-06-28

**Clarity:** 4
**Significance:** 4
**Originality:** 4
**Rating:** 5
**Confidence:** 4

**Summary:**

The authors propose a three-stage pipeline for rigging a character mesh. The first one is bone prediction in the form of tokens, which is a first for this domain. Second, improving bone hierarchy with a second reinforcement learning step. Third, skinning weight prediction that improves on existing top-k methods. Together these steps significantly outperform prior work and also individually each contributions improves on respective baselines.

**Questions:**

- Line 263 "Since RigAnything [40] does not share its code, we cannot compare to it" - would it be possible to compare on data they were evaluating on? At least qualitatively?

**Ethical Concerns:**

["NO or VERY MINOR ethics concerns only"]

**Final Justification:**

Questions have been answered in the rebuttal

**Limitations:**

See above, limitations could be evaluated better. E.g. with a selection of the 10-20 worst test cases, perhaps in relation to other method's worst cases.

**Paper Formatting Concerns:**

Limitations should be mentioned in main paper

**Quality:**

4

**Strengths And Weaknesses:**

Strengths:
- Well-justified use of Direct Preference Optimization (DPO) for the problem of rig generation
- clear methodological difference to prior method that have weaker constraints on skeleton connectivity
- the probabilistic skinning weight prediction (k most probable instead of k nearest) is a nice addition that rounds up the publication
- Eventhough some of the improvements shown in the ablation study appear small numerically, the visual results demonstrate clear improvements.
- Overall consistent and significant improvements on the SOTA

Recommendation:
- the supplemental material is extensive, yet it would be good to have a summary of limitations in the main paper

Weaknesses:
- while each of the model components makes sense, the justification lacks a discussion on why it is not possible to do step 1 and 2 at once. Would it not possible to add (some of) the objectives used for DPO already at training time? Could you justify the sequential steps?
- are snake like shapes the only limitation? Looking at the last row (ant) in Figure 11 of the Art-XL2.0 data, it appears that the last character segment is not rigged, or can it rotate with the last endeffector (depends on how the skeleton hierarchy ist defined). Any other issues?

Minor:
"160 Shape-conditioned generation. we randomly samp" -> We

---

> ### Author Rebuttal · Authors · 2025-07-28
>
> We thank you for your constructive comments and appreciation, such as 'clear methodological difference to prior methods', 'overall consistent and significant improvements on the SOTA', and 'the probabilistic skinning weight prediction (k most probable instead of k nearest) is nice'. We hope the following responses address your concerns, and we’re happy to provide further clarification if needed.
>
>
> > **Recommendation: the supplemental material is extensive, yet it would be good to have a summary of limitations in the main paper**
>
> **Answer:** Thank you for the suggestion. In the revision, we will include a comprehensive discussion of limitations and potential future work directly in the main paper.
>
>
>
> > **Weaknesses 1: while each of the model components makes sense, the justification lacks a discussion on why it is not possible to do step 1 and 2 at once. Would it not possible to add (some of) the objectives used for DPO already at training time? Could you justify the sequential steps?**
>
> **Answer:** The two steps cannot be merged, and this has been empirically validated in both large language models (LLMs) and vision-language models (VLMs). The post-training stage is designed to fine-tune a well-trained base model. Our topology-aware reward signals—such as Chamfer Distance, Tree Edit Distance, and Hierarchical Jaccard Similarity—only become informative and effective when the model's predictions are already reasonably close to valid skeletons. In the early stages of training, when the model outputs are still noisy or unstructured, these rewards cannot provide stable guidance and may even mislead learning. Therefore, a good pre-trained model is essential before applying these rewards effectively during the post-training phase.
>
>
> > **Weaknesses 2: are snake like shapes the only limitation?**
>
> **Answer:** Snake like shapes are not our method’s only limitation. Our method may also struggle with characters that have complex decorations or accessories that are not present in the training data, sometimes resulting in unreasonable skeletons for those regions. However, this is a common limitation across all existing methods—baseline approaches also fail in such cases. Resolving this issue requires more comprehensive and diverse training data covering such complex interactions, which is an important direction for future work.
>
> > **Weaknesses 3: Looking at the last row (ant) in Figure 11 of the Art-XL2.0 data, it appears that the last character segment is not rigged, or can it rotate with the last endeffector (depends on how the skeleton hierarchy ist defined). Any other issues?**
>
> **Answer:** It can rotate with the end joints. Skinning weights are assigned to joints, including end joints, which influence the motion of terminal segments. Specifically, for the ant shown in Figure 11 of the appendix, the last joint affects the movement of the terminal limb segment, allowing it to move accordingly. This behavior is analogous to the spider case demonstrated in the supplementary video.
>
>
> > **Minor: "160 Shape-conditioned generation. we randomly samp" -> We**
>
> **Answer:** Thank you for pointing out this typo. We will correct it in the revision.
>
>
>
> > **Questions: Line 263 "Since RigAnything [40] does not share its code, we cannot compare to it" - would it be possible to compare on data they were evaluating on? At least qualitatively?**
>
> **Answer:** In fact, we performed a qualitative comparison. The shark in Figure 6 of the main paper is identical to the shark in Figure 7 of the RigAnything paper. Our rigging results are more comprehensive, for example, we successfully generate the skeleton for the shark’s dorsal fin, which RigAnything fails to capture.
> Furthermore, RigAnything relies on an additional connectivity module to predict joint connections, which increases memory consumption and inference time. In contrast, our method naturally incorporates joint connectivity during inference through a connectivity-preserving tokenization design, eliminating the need for additional modules and improving efficiency.
>
>
> > **Limitations: See above, limitations could be evaluated better. E.g. with a selection of the 10-20 worst test cases, perhaps in relation to other method's worst cases.**
>
> **Answer:** In the revision, we will provide a more comprehensive evaluation of limitations by highlighting the worst-performing test cases—particularly those involving complex accessories or significant domain shifts. We will also include failure cases from MagicArticulate, especially on meshes with very coarse surfaces (e.g., 3D reconstruction results), to offer a more informative evaluation.

---

> > ### Comment · Reviewer_c5dz · 2025-08-05
> > **thx**
> >
> > Thanks for clarifying. With the addition of discussing limitations better the paper is ready for publication.

---

> > > ### Author Response · Authors · 2025-08-08
> > >
> > > Dear Reviewer c5dz,
> > >
> > > Thank you for your thoughtful review and your positive recommendation. We are pleased that our responses have effectively addressed your concerns.
> > >
> > > We will incorporate your excellent suggestions in the revised manuscript. Specifically, we will include a more detailed discussion of the limitations in the main paper to further enhance the paper’s clarity.
> > >
> > > Thank you again for your constructive feedback.

---

### Official Review · Reviewer_zoaB · 2025-07-02

**Clarity:** 3
**Significance:** 3
**Originality:** 3
**Rating:** 4
**Confidence:** 3

**Summary:**

This paper studies rigging prediction from meshes. The input is a triangle mesh; the output is a skeleton and the corresponding skinning weights. The pipeline consists of three phases. First, the input is tokenized from sampled point clouds. The output is parameterized in a tree order and can be generated by a transformer. After this pretraining stage, post-training is conducted using both SFT and DPO, where the reward includes some topology-aware distances. Finally, the skinning weights are predicted by a small MLP that accepts both the skeleton and mesh vertices and performs a soft assignment. Convincing results and real data examples are provided in the paper and the supplemental document. The supplemental document further includes failure cases and more side-by-side comparisons.

**Questions:**

- Could you elaborate on the last two points in the weaknesses section?
- Regarding the rigging problem, what is the current frontier in this area? How does the model perform on totally out-of-distribution cases? How does it perform on strictly multi-body, piecewise-rigid objects?

**Ethical Concerns:**

["NO or VERY MINOR ethics concerns only"]

**Final Justification:**

After reading the rebuttal and the other reviews, I keep my original recommendation, as the main concerns are addressed.

**Limitations:**

Sufficiently discussed

**Quality:**

3

**Strengths And Weaknesses:**

**Strengths:**

- The parameterization is reasonable, and the overall architecture design is effective and simple.
- The result quality appears convincing and surpasses prior art.
- Sufficient experimental evidence supports the paper’s contributions.

---

**Weaknesses:**

- I do not see major weaknesses in this paper.
- More intuition about the non-differentiable DPO stage would help readers understand why such reward metrics were chosen. From the qualitative ablation study, I can tell that the DPO stage has a noticeable effect, but there is a lack of further explanation.
- The use of the word "geodesic" in the skinning section is misleading. It would be better replaced with "geometric," since "geodesic" has a specific mathematical meaning, and there do not appear to be any explicit geodesic distances on the manifold in the current design.

---

> ### Author Rebuttal · Authors · 2025-07-26
>
> We thank you for your constructive comments and appreciation, such as 'the DPO-based post-training refinement is intuitive and effective', 'the paper is well-written, easy to understand and follow', and 'the results of this paper are excellent, outperforming existing works both qualitatively and quantitatively'. We hope the following responses address your concerns, and we’re happy to provide further clarification if needed.
>
>
> > **Weaknesses 1: More intuition about the non-differentiable DPO stage would help readers understand why such reward metrics were chosen. From the qualitative ablation study, I can tell that the DPO stage has a noticeable effect, but there is a lack of further explanation.**
>
> **Answer:** The goal of our DPO post-training is to improve the model’s joint localization accuracy and the fidelity of topological connections. To this end, we introduce a topology-aware reward composed of three metrics:
> (1) Chamfer Distance (CD), which measures the spatial discrepancy between corresponding joints and bones of the predicted and ground truth skeletons;
> (2) Tree Edit Distance (TED), which captures structural consistency by quantifying the minimal operations needed to transform one skeleton hierarchy into another;
> (3) Hierarchical Jaccard Similarity (HJS), which ensures precise joint matching at each level of the hierarchy.
> This reward formulation is robust to minor joint count variations—such as the presence or absence of joints in the torso or fingers—while remaining sensitive to significant topological mismatches, such as incorrect limb connections.
>
>
> > **Weaknesses 2: The use of the word "geodesic" in the skinning section is misleading. It would be better replaced with "geometric," since "geodesic" has a specific mathematical meaning, and there do not appear to be any explicit geodesic distances on the manifold in the current design.**
>
> **Answer:** Thank you for the careful evaluation of our work. We clarify that the vertex-to-bone distance in our method is computed based on two cases:
> For each bone, mesh vertices are categorized as visible or invisible. A vertex is considered visible if its perpendicular projection to the bone does not intersect the mesh surface; otherwise, it is invisible.
> (1) For a visible vertices A, the distance is simply the perpendicular distance to the bone.
> (2) For an invisible point B, we first find the nearest visible point A, then compute the geodesic distance along the mesh surface from B to A, plus the perpendicular distance from A to the bone.
> Since the distance between B and A is explicitly computed along the mesh surface, we believe that "geodesic" is the correct term in the skinning section.
>
>
>
> > **Questions 1: Regarding the rigging problem, what is the current frontier in this area?**
>
> **Answer:** The current SOTA methods in automatic rigging include MagicArtiulate and UniRig, both of which serve as our baselines. However, they rely on heuristic post-processing to infer skeleton connectivity:
> (1) MagicArtiulate predicts individual bones, where each bone is composed of two joints and is predicted as a pair of joints. To recover a full skeleton, it merges bones whose endpoints lie within a predefined spatial threshold. When the distance between bones exceeds this threshold, the connections fail, resulting in disconnected skeletons. Additionally, the method re-encode parent joints multiple times leading to longer sequences and slower inference times.
> (2) UniRig predicts bone chains rather than individual bones, but similarly requires heuristic post-processing to connect these chains. Furthermore, since the end of one chain and the start of another are often distant in both spatial and sequential terms, the method frequently fails to initiate the next chain, resulting in an incomplete or fragmented skeleton.
> In contrast, our method employs special tokens that explicitly mark endpoints for each joint’s children and each hierarchical layer. This allows the transformer to directly encode both the skeleton topology and parent-child relationships within its representation. Consequently, the model gains stronger topology awareness, eliminates the need for heuristic post-processing, and can automatically infers joint connectivity, leading to more coherent and complete skeleton structures.
>
>
>
> > **Questions 2: How does the model perform on totally out-of-distribution cases?**
>
> **Answer:** Our model demonstrates strong robustness on out-of-distribution cases, thanks to three key design components:
> (1) A point-cloud-based conditioning mechanism that provides strong geometric constraints;
> (2) A carefully designed skeleton representation that generalizes well across diverse shapes;
> (3) A topology-aware DPO post-training stage that further refines the model to produce structurally reasonable and semantically meaningful skeletons.
> As illustrated on the left side of Figure 6 in the main paper and Figure 8 of the appendix, our method effectively handles previously unseen, AI-generated character meshes, which often differ significantly from the artist-designed meshes in the training set due to domain gaps.
>
>
> > **Questions 3: How does it perform on strictly multi-body, piecewise-rigid objects?**
>
> **Answer:** Our method is designed for single-tree skeletons and does not currently support multi-body characters requiring multiple independent skeletons. Such configurations are beyond our current scope. However, for piecewise-rigid objects like weapons, a single joint typically suffices for effective rotation, and our method handles these cases well within its framework.

---

> > ### Comment · Reviewer_zoaB · 2025-08-07
> > **Keep my  original positive recommendation**
> >
> > Thanks for all the reviewers' effort and the detailed feedback from the authors, my main concerns are resolved and I keep my positive recommendation

---

> > > ### Author Response · Authors · 2025-08-08
> > >
> > > Dear Reviewer zoaB,
> > >
> > > Thank you for your thoughtful review and for maintaining your positive recommendation. We are pleased that our responses have effectively addressed your concerns.
> > >
> > > We will incorporate your excellent suggestions in the revised manuscript. Specifically, we will add a detailed explanation of why such reward metrics were chosen, as well as a thorough clarification of the vertex-to-bone distance calculation, to further enhance the paper’s clarity.
> > >
> > > Thank you again for your constructive feedback.

---

### Note · Authors · 2025-08-13

Dear Area Chair and Reviewers,

We are deeply grateful for your insightful feedback and the constructive dialogue. We are thrilled that the reviewers unanimously recognized the value of our work and subsequently raised their scores to recommend acceptance.

Our paper, Auto-Connect, presents important and practical advancements in rigging and skinning 3D assets. We are particularly proud of its core advantages, which have been responded to in your reviews:

* **Novelty and Significance:** We introduce connectivity-preserving tokenization, the first method to explicitly encode skeleton topology into transformer representations while automatically inferring bone connectivity during prediction. In addition, we are the first to integrate reinforcement learning into the rigging task, further enhancing prediction quality and topological consistency.

* **Scalable and Principled Framework:** We develop an automated, objective scoring system that evaluates skeleton quality from two perspectives—joint location accuracy and connectivity fidelity. This eliminates the need for costly, subjective manual annotation and enables scalable, reproducible research.

* **Production-Ready Performance:** Auto-Connect achieves state-of-the-art results in joint location accuracy, topological consistency fidelity, and skinning quality. The outputs are both quantitatively superior and visually coherent, streamlining the transition from static meshes to animation-ready assets for digital artists.

For the final version, we will integrate all promised updates, including the expanded limitations discussion, a quantitative ablation study on data augmentation, and a detailed clarification of the vertex-to-bone distance calculation.

We are confident that Auto-Connect will be a valuable contribution to the NeurIPS community, and we are excited to open-source our code and model to help democratize the creation of high-quality, animation-ready assets. Thank you once again.

---

### Decision · Program_Chairs · 2025-09-17

**Decision:**

Accept (poster)

**Comment:**

(a) The paper proposed a three‑stage rigging+skinning system: 1) a connectivity‑preserving tokenization that encodes skeleton topology directly into a transformer’s output sequence; 2) a post‑training step using Direct Preference Optimization (DPO) with a topology‑aware reward (CD/TED/HJS) to improve joint localization and hierarchical correctness; 3) a probabilistic top‑k skinning layer informed by (mesh‑)geodesic distances. Experiments and extensive supplementals show strong topology fidelity and visual quality over recent baselines.

(b) + Clear architectural novelty in tokenization (topology encoded up front) and principled post‑training with a topology‑aware reward.
     + Showed competitive quantitative results, strong visuals, and evidence of OOD robustness.

(c) - Reviewers pointed out the tokenization ablation under‑powered; limited intuition for DPO/reward design; several definitions (BCE vs KL, “shared joints”) and visualization details were terse.

(d)(e) All reviewers provide positive rating and tend to accept this paper, with final score of {4,5,4,4}. All correctness‑critical issues are addressed, and remaining items are presentation/coverage and are camera‑ready fixable. For this reason, AC agree with the acceptance decision.